# Prevalence of Mismatch Repair Deficiency in Advanced Solid Tumors (Colorectal Cancer and Non-Colorectal Cancer) in One Mexican Institution

**DOI:** 10.3390/jpm14121152

**Published:** 2024-12-13

**Authors:** Rita Dorantes-Heredia, Daniel Motola-Kuba, Ixel Escamilla-López, Eduardo Téllez-Bernal, Emilio Conde-Flores, Daniel Escalera-Santamaría, Emilio Medina-Ceballos, José Ruiz-Morales, Elena Dorokhova, Lucia Edith Flores-García, Gabriela Lugo, Georgina del C. Filio-Rodríguez

**Affiliations:** 1Departamento de Anatomía Patológica, Médica Sur, Mexico City 14050, Mexico; rdorantesh@medicasur.org.mx; 2Centro Oncológico, Médica Sur, Mexico City 14050, Mexico; dmotolak@medicasur.org.mx (D.M.-K.); econdef@medicasur.org.mx (E.C.-F.); escalerasantamaria31@gmail.com (D.E.-S.); emilio_medina94@hotmail.com (E.M.-C.); mruizm@medicasur.org.mx (J.R.-M.); 3Unidad Médica Onco-Hematológica de Puebla, Puebla 72424, Mexico; ixel.escamilla@gmail.com (I.E.-L.); umoetb@yahoo.com (E.T.-B.); 4Global Medical and Scientific Affairs, MSD, Mexico City 01090, Mexico; elena.dorokhova@merck.com (E.D.); lucia.flores@merck.com (L.E.F.-G.)

**Keywords:** colorectal cancer (CRC), microsatellite instability, deficient mismatch repair (dMMR)

## Abstract

**Background/Objectives**: Mismatch repair (MMR) status is an important prognostic and predictive indicator in cancer, distinguishing proficient (pMMR) tumors from deficient (dMMR) ones. This study aimed to determine the prevalence of dMMR in colorectal (CRC) and selected non-CRC solid tumors (gastric, esophageal, and endometrial cancers). **Methods**: This retrospective study was conducted at a private health institution in Mexico City, analyzing patients diagnosed with colorectal, gastric, esophageal, or endometrial cancer from January 2017 to December 2020. dMMR prevalence was assessed using available status information and tissue samples for immunohistochemistry (IHC). Data were analyzed via SPSS, presenting results in frequencies and percentages. **Results**: Most solid tumors exhibited MSH2, MSH6, and MLH1 expression above 90%, with slightly lower levels in endometrial cancer. Esophageal cancer showed 100% pMMR. dMMR prevalence was found to be 12.7% for CRC, 8.3% for gastric, and 18.5% for endometrial cancers. Prevalence rates were similar across genders (11.1% in women and 12.9% in men), with the highest prevalence in the 41–50 age group (20%) and the lowest in the 31–40 age group (7.7%). **Conclusions**: This study offers valuable insights into the frequency of dMMR mutations in a cohort of the Mexican population, providing a basis for further research on their prevalence in Mexico.

## 1. Introduction

One simple and useful prognostic and predictive indicator in cancer is whether a tumor is proficient (pMMR) or deficient (dMMR) for mismatch repair (MMR); multiple investigations have shown that the dMMR subgroup has a better prognosis [1]. Mismatch repair deficiency usually occurs due to mutations that code for genes of mismatch repair (MMR) proteins (MLH1, MSH2, MSH6, and PMS2) that are responsible for recognizing and correcting errors in mismatched nucleotides or through methylation of the MLH1 gene promoter. These errors in MMR lead to microsatellite instability (MSI) due to the accumulation of errors in DNA microsatellites (short repetitive sequences in DNA) [2].

Many primary malignancies are MSI-H or dMMR [3]. MSI-H tumors have lymphocytic and other immune cell infiltration, medullary histology, poorly differentiated histology, as well as a hypermutated phenotype [4]. To identify dMMR tumors, simple tests such as MSI testing [5] and immunohistochemistry (IHC) testing for MMR proteins are frequently applied [6]. MSI-H or dMMR occurs at a rate of 26% for endometrial cancer, 10% for colorectal carcinoma, and 8% for gastric cancer [7]. The metastatic colorectal tumor appears to have a lower rate of 5% MSI-H [3].

In the largest real-world Asian MSI dataset (26,237 samples), 3.72% of patients with unresectable or metastatic solid tumors had MSI-H. MSI-H was 1.8-fold higher in female patients (4.75%) than male patients (2.62%), but excluding endometrial cancer, it was 3.2%, similar to male patients [8].

In a study carried out at the National Cancer Institute in Mexico, in which the determination of dMMR with immunohistochemistry was performed prospectively for 202 young patients who presented consecutively with colorectal cancer (CRC) for the first time at different stages of the disease, 43 (21.3%) cases showed dMMR. The only clinicopathological characteristics associated with dMMR were its location in the right colon and the presence of a family history of cancer. In the multivariate analysis, the presence of the tumor in the right colon was associated with dMMR (OR = 5.823, 95%-C.I. = 2.653–12.784, *p* < 0.001) [9]. In another study, the objective of which was to assess MMR abnormalities in the tumors of Mexican CRC patients under 50 years old, CRC paraffin-embedded tissues of 47 patients with available demographic/clinical data were studied by immunohistochemistry (IHC) for MLH1/MSH2, qPCR with specific probes/sequencing for the BRAF V600E mutation, and conventional PCR (5 markers) for MSI analysis. The study found 42.6% of cases with defective MMR expression [10]. In the study of Luévano-González et al., the aim of which was to determine if age is a risk factor for defective MMR protein expression and BRAF mutations in their population, they found an association between young age and defective MMR expression [11]. Currently, MSI-H, or Loss of mismatch repair protein expression, along with PD-L1 expression, are biomarkers used to narrow down those patients expected to respond to immune checkpoint inhibitors [12].

On 23 May 2017, the FDA approved pembrolizumab for the treatment of adult and pediatric patients with unresectable or metastatic MSI-H or dMMR solid tumors that have progressed after prior treatment and have no satisfactory alternative treatment options, and for the treatment of unresectable or metastatic colorectal cancer that has progressed after treatment with a fluoropyrimidine, oxaliplatin, and irinotecan [13]. On 29 June 2020, the FDA approved pembrolizumab for first-line therapy of unresectable MSI-H or dMMR colorectal cancer [14]. This is the first time the agency has approved a cancer treatment based on a common biomarker rather than an organ-based approach [15], and the precedent has been set for this all-important new approach.

The present study aims to determine the prevalence of dMMR in selected solid tumors (CRC and the more common non-CRC cohort [gastric, esophageal, and endometrial cancer]) in the study population in terms of percentage and/or frequency by tumor type and patient characteristic (age, gender, initial tumor/metastasis) and to describe the biomarker profile of colorectal cancer patients (dMMR, KRAS, NRAS, BRAF).

## 2. Methods

This is a retrospective data-based study that used data gathered from a private health institution in Mexico City, Médica Sur, a member of the Mayo Clinic Care Network, including 215 (*n* = 215) records from institution patients. The database was completely electronic. All patients diagnosed with colorectal, gastric, esophageal, or endometrial cancer between 1 January 2017 and 31 December 2020 and treated at Médica Sur were included in the study. Initially, only 65 records had dMMR status information, and to increase the sample size, the pathology department carried out the analysis of approximately 150 more tissue samples stored at the site for IHC with the same commercial kit.

This study used secondary data sources and did not involve human subjects. The database did not include sensitive information that could lead to the identification of patients. The collection and analysis of data from participating patients were limited to those indicated in the protocol. No personal data were shared with the sponsor or with the alternative contacts, so there was no need to obtain informed consent. As this study was carried out in Mexico, approval from an Institutional Review Board/Independent Ethics Committee (IRB/IEC) was required and obtained prior to the study’s start.

### 2.1. Statistical Analysis

The initial descriptive analysis serves to describe the study’s characteristics. The prevalence of dMMR in selected solid tumors in the sample was calculated and expressed in frequencies and percentages. The biomarker profile of colorectal cancer patients (dMMR, KRAS, NRAS, BRAF) was expressed in frequencies and percentages. The prevalence of dMMR was described per tumor type (primary tumor/metastasis) and patient characteristics (including demographics and disease information) in frequencies and percentages. No hazard ratios were determined in the present analysis. Statistical analysis was conducted in SPSS version 29.0.1.0.

### 2.2. Immunohistochemistry

Formalin-fixed and paraffin-embedded whole tissue sections, each measuring 4 µm in thickness, were prepared for immunohistochemical analysis. The immunohistochemistry (IHC) was performed using the following antibodies: MLH1 (G168-728), MSH2 (G219-1129), MSH6 (clone EP49), and PMS2 (EP51). Immunostains were performed using the standard avidin-biotin peroxidase method described in Table 1; all were studied with the appropriate controls. The control tissue used was normal colon.

The resulting immunohistochemical slides were evaluated by two independent pathologists, who were blinded to each other’s assessments and had no prior knowledge of the clinicopathological parameters. Additionally, a third pathologist reviewed the pathology reports. In cases of discrepancies (which occurred in less than 5% of instances), a joint observation was conducted to reach a conclusive agreement.

Loss of MMR (mismatch repair) protein expression was defined by the absence of immunohistochemical staining in the nuclei of neoplastic cells. Cases were stratified into two distinct groups based on MMR protein expression following the recommendations of the College of American Pathologists (CAP):Group 1: Proficient MMR (pMMR): Defined as cases with retained expression of all four MMR proteins.Group 2: Deficient MMR (dMMR): Defined as cases with Loss of expression of one or more proteins, specifically within the MLH1/PMS2 and/or MSH2/MSH6 pairs. The classification of reduced MMR (low-patchy MMR) was not considered in this analysis, considering this classification as not widely accepted by CAP [14].

## 3. Results

The clinical characteristics of patients that met the inclusion criteria are described in Table 2.

Initially, the immunohistochemistry (IHC) testing for mismatch repair (MMR) proteins according to each type of solid tumor is shown in the table below.

In all types of solid tumors, intact nuclear expression remained above 90%, except for endometrial cancer, where the intact protein expression of MSH2, MSH6, and MLH1 was slightly below. It is interesting to note that the expression of the four proteins was reported to be intact in all cases of esophageal cancer (Table 3).

Table 4 shows the prevalence of dMMR in the selected solid tumors (CRC, gastric, esophageal, and endometrial cancer) in the population of the study.

In accordance with the data in the table above, 100% of esophageal cancers are pMMR, with the rest as dMMR in 12.7%, 8.3%, and 18.5% for CRC, gastric, and endometrial cancers, respectively.

In the biomarkers profile of colorectal cancer patients (dMMR, KRAS, NRAS, BRAF) in the population of the study, the only biomarker used was KRAS, which was not routinely available and was only given to a few patients through a commercial program unrelated to the researchers; therefore, not all exploratory outcomes were reached, and some results were inconclusive.

The KRAS biomarker was found to be mutated in 3.33% of CRC patients, non-mutated or wild type in 7.33%, and unknown in the remaining 89.33%.

Finally, the tables below describe the prevalence of dMMR in the study population based on tumor type and patient characteristics (gender, age, primary tumor/metastasis).

In the case of the results obtained for the prevalence of dMMR based on gender (Table 5), similar total values were found: 11.1% in women and 12.9% in men, where the lowest percentage for women was found in CRC, which was 8.3%, unlike men, with about double, at 15.6%. Endometrial cancer had the highest percentage of dMMR in women, taking account for 18.5% of cases.

In the comparative case of dMMR corresponding to the analysis site (Table 6), final percentages were found with 13.6% in the case of metastatic cases and 11.9% in primary tumors. In the case of gastric cancer, where metastatic, dMMR was reported in 25%, unlike primary tumors, with 5% reported, and unlike endometrial cancer, where dMMR was reported in the metastatic case in 0% and in the case of the primary tumor at 20%.

In terms of dMMR by age group (Table 7), it was seen that the highest total percentage was found in the group of 41–50 years, with 20%, without considering that in the three cases of CRC in those under 30 years of age, one was reported as dMMR. The age group with the lowest percentage, on the other hand, was 31–40, with 7.7%. It is interesting to note that in the three cases of patients over 91 years of age, they were CRC pMMR.

## 4. Discussion

This study provides insights regarding the expression of mismatch repair proteins (MMR) in four solid tumors classified in a CRC cohort and a non-CRC cohort (including stomach, esophageal, and endometrial cancer).

In the initial results of the immunohistochemistry (IHC) tests for mismatch repair proteins (MMR) according to each type of solid tumor, it is noteworthy that in all cases (MSH2, MSH6, MLH1, PMS2), they remained intact in esophageal cancer, as well as in gastric cancer for MSH2 and MSH6. In the remaining cases, intact nuclear expression remained greater than 85%.

In the context of colorectal cancer, the national literature indicates that the prevalence of deficient mismatch repair (dMMR) typically ranges from 21% to 34%. This figure is notably higher than the 12.7% reported in our study. In a study focused on Latin populations, the prevalence of dMMR tumors approached 13% [1]. Some authors, including Gutierrez and colleagues, have identified factors such as ethnicity and previous cancer diagnoses as associated with an increase in microsatellite instability-high (MSI-H)/dMMR cases [16]. Additionally, our findings reveal that the rate of dMMR is higher in men compared to women, with a prevalence of 15.6% (*n* = 14/90) in men versus 8.3% (*n* = 5/60) in women. This observation is noteworthy, given that most of the literature suggests a predominance of dMMR in women [3].

The KRAS biomarker has limited availability, and it was only provided to a small number of patients through a commercial program that was not affiliated with the researchers. Despite the limited number of tests conducted, KRAS mutations were detected in 3.33% (*n* = 5) of colorectal cancer (CRC) patients, while 7.33% exhibited the non-mutated or wild-type form, leaving 89.33% (*n* = 135) of cases classified as unknown. Values were close to what was demonstrated in a CRC study in Latin American patients, where 30% of patients (85 patients) had the KRAS test and 40.0% (34 patients) had KRAS mutant cancers [1].

In the context of solid tumor analysis, the incidence of dMMR was relatively comparable between male and female patients, with approximately 12.9% of males and 11.1% of females exhibiting dMMR. Notably, a small disparity was observed, as the lowest percentage of dMMR among women was found in colorectal cancer (CRC), where it was reported at 8.3%. In contrast, the prevalence of dMMR in men diagnosed with CRC was approximately double, at 15.6%. Endometrial cancer demonstrated the highest prevalence of dMMR in female patients, accounting for 18.5% of cases. This finding aligns with the results reported by Tetsuya-Ito et al., who identified a dMMR prevalence of 17.2% (68 out of 395 cases) in their study [17].

For patients with unresectable or metastatic solid tumors, in this study, a total of 13.6% dMMR was found in a metastatic setting and 11.9% in primary tumors, for a total of 12.1%, a percentage close to the 16% reported in the international literature for overall weighted dMMR prevalence across all tumor types and stages after a sensitivity analysis, where specifically dMMR all-stage prevalence for the United States was estimated at 14% and for Japan was estimated at 20% [4].

In this study, dMMR was found in 18.5% of endometrial cancer cases, 12.7% of colon cancer cases, and 8.3% of gastric cancer cases. However, it was not detected in any cases of esophageal cancer, as expected from the previous results of IHC tests for MMR.

In all cases, except for endometrial cancer dMMR, in which the pooled prevalence from 26 studies (5248 patients) was estimated at 25% (22–28%), which is a little higher than the one reported in this study, the rest shows remarkable consistency with the international data, in which, for CRC, the dMMR pooled prevalence from 4 studies (11,434 patients) was estimated at 10% (5–15%), gastric cancer dMMR pooled prevalence from 4 studies (854 patients) was estimated at 8% (2–17%), and dMMR analysis was not feasible for esophageal tumors [4].

However, the results of this study revealed variations compared to the existing literature when analyzing the incidence of MSI-H or dMMR based on tumor stage. For endometrial cancer, previous studies have reported an incidence of dMMR in patients undergoing curative resection as high as approximately 30%. In contrast, our findings indicated a dMMR prevalence of 20% in primary tumors, with no cases (0%) observed in metastatic cases.

For gastric cancer, a meta-analysis conducted by Lorenzi and Cols suggested that dMMR occurs in approximately 11% (9–12%); independently of the stage, and based on stages across gastrointestinal tumors, the prevalence was 13% (10–16%; 10 studies; 3194 patients) for stages 1–2, and the prevalence was 10% (7–13%; 10 studies; 1319 patients) in stages 3–4 cancer. The highest MSI-H pooled prevalence was observed for the intestinal histological subtype, at 13% (10–17%). This contrasts with our findings in gastrointestinal cancer, where we observed a dMMR prevalence of 20%. Additionally, for most other tumor types, dMMR has been generally reported at less than 5% [7]. In this study, we documented dMMR rates of 12.7%, 5%, and 0% for colorectal, gastric, and esophageal cancers, respectively.

In terms of dMMR by age group, it was seen that the highest total percentage was found in the 41–50 years group with 20%, without considering that in the three cases of CRC in those under 30 years of age, one was reported as dMMR. The age group with the lowest percentage, on the other hand, was 31–40, with 7.7%. It is interesting to note that in the three cases of patients over 91 years of age, they had CRC pMMR, and in patients > 60 years old, dMMR was exclusively presented in CRC with the exception of a small group of endometrial cancer patients 61–70 years old, with 21.4% of dMMR (*n* = 3/14).

With the completion of this study, important information on the frequency of these mutations in the Mexican population was provided, laying the groundwork for larger studies to gain a better understanding of their prevalence in Mexico, as the results can differ drastically depending on a cohort. Because of recent evidence supporting the role of MSI-H/dMMR and associated immunogenicity as a mechanism for increased efficacy of the anti-PD-1 immune checkpoint monoclonal antibody pembrolizumab for MSI-H or dMMR in unresectable or metastatic tumors with MSI-H or dMMR, regardless of age or histotype, this can help pave the way for future clinical research into the behavior of MSI-H/dMMR and its therapy to lead to a significantly longer progression-free survival (PFS) with fewer treatment-related adverse events.

### 4.1. Future Directions of the Research

Genetic Variability and Ethnicity: The study provided valuable insights into the prevalence of dMMR in a reduced cohort of the Mexican population. However, future research should focus on investigating the genetic variability of MMR genes and their association with dMMR across different ethnic groups and the wider Mexican population. Understanding genetic differences in dMMR prevalence among diverse populations can help in tailoring effective treatment strategies for specific ethnicities [16].Biomarker Profiling and Treatment Response: Further research is needed to explore the interplay between dMMR status, KRAS, NRAS, and BRAF mutations and treatment response. Investigating the impact of these biomarkers on the efficacy of checkpoint inhibitors, such as pembrolizumab, in patients with dMMR solid tumors can provide valuable insights into personalized treatment approaches [17].Long-Term Clinical Outcomes: It is essential to conduct longitudinal studies to assess the long-term clinical outcomes of patients with dMMR solid tumors. Research should focus on evaluating overall survival, disease-free survival, and treatment-related adverse events in relation to dMMR status. Longitudinal studies can also shed light on the impact of dMMR on the progression-free survival (PFS) of patients undergoing different treatment modalities.Age-Related Patterns of dMMR: The study highlighted associations between age groups and dMMR prevalence. Future research can delve deeper into age-related patterns of dMMR across different cancer types. Understanding the influence of age on dMMR status can aid in the development of age-specific screening and treatment protocols [18].Immunogenomic Profiling: Given the association between dMMR and immune cell infiltration, future research should focus on the immunogenomic profiling of dMMR tumors. Investigating the tumor microenvironment, immune cell subtypes, and their interaction with dMMR tumors can provide crucial insights into the immune response and potential immunotherapeutic targets.MSI-H and PD-L1 Expression: Further studies are warranted to understand the correlation between MSI-H status, PD-L1 expression, and response to immune checkpoint inhibitors. Exploring the predictive value of combined MSI-H and PD-L1 expression in determining the response to immunotherapy can guide precision medicine in the treatment of dMMR solid tumors [19].Real-World Data Validation: It is important to validate the findings of this study using real-world data from larger cohorts. Collaborative efforts involving multiple healthcare institutions and diverse patient populations can provide a comprehensive understanding of dMMR prevalence, biomarker profiles, and treatment outcomes [20].

These future research directions aim to expand the current knowledge on dMMR in solid tumors, paving the way for personalized and effective therapeutic interventions across diverse patient populations.

### 4.2. Limitations

As this was a descriptive study of a non-randomized sample of solid tumor patients from a database from a private health institution and no inferential statistical analysis was conducted, no conclusions regarding causality can be drawn directly from this work.

Preanalytical issues related to tissue fixing and care, and a lack of standardization of the molecular techniques, can represent a bias in the results reported in the database.

The following limitations potentially affect the generalizability of findings or interpretation of the results:Data reflect limited data from patients seen and treated.Data may not be representative of the whole country.It is likely to have missing data.

## Figures and Tables

**Table 1 jpm-14-01152-t001:** Antibodies and dilutions used in the immunohistochemistry assay.

Antibody	Clone	Dilution
MLH1	G168-728	1:100
MSH2	G219-1129	1:100
MSH6	EP49	1:100
PMS2	EP51	1:50

**Table 2 jpm-14-01152-t002:** Clinical characteristics.

		Colorectal		Gastric		Endometrial		Esophageal	
		*n*	*n* (%)	*n*	*n* (%)	*n*	% *n* (%)	*n*	*n* (%)
Sex	Male	90	60.0%	14	58.3%	0	0.0%	12	85.7%
	Female	60	40.0%	10	41.7%	27	100.0%	2	14.3%
	Total	150	100.0%	24	100.0%	27	100.0%	14	100.0%
Age	<30	3	2.0%	0	0.0%	0	0.0%	0	0.0%
	31–40	9	6.0%	3	12.5%	0	0.0%	1	7.1%
	41–50	14	9.3%	0	0.0%	4	14.8%	2	14.3%
	51–60	27	18.0%	7	29.2%	7	25.9%	7	50.0%
	61–70	41	27.3%	7	29.2%	14	51.9%	3	21.4%
	71–80	43	28.7%	4	16.7%	2	7.4%	1	7.1%
	81–90	10	6.7%	3	12.5%	0	0.0%	0	0.0%
	>91	3	2.0%	0	0.0%	0	0.0%	0	0.0%
	Total	150	100.0%	24	100.0%	27	100.0%	14	100.0%
Site of Analysis	Primary	134	89.3%	20	83.3%	25	92.6%	14	100.0%
	Metastasis	16	10.7%	4	16.7%	2	7.4%	0	0.0%
	Total	150	100.0%	24	100.0%	27	100.0%	14	100.0%
MSH 2	Loss of nuclear expression	3	2.0%	0	0.0%	3	11.1%	0	0.0%
	Intact nuclear expression	147	98.0%	24	100.0%	24	88.9%	14	100.0%
	Total	150	100.0%	24	100.0%	27	100.0%	14	100.0%
MSH6	Loss of nuclear expression	7	4.7%	0	0.0%	3	11.1%	0	0.0%
	Intact nuclear expression	143	95.3%	24	100.0%	24	88.9%	14	100.0%
	Total	150	100.0%	24	100.0%	27	100.0%	14	100.0%
MLH1	Loss of nuclear expression	15	10.0%	2	8.3%	3	11.1%	0	0.0%
	Intact nuclear expresion	135	90.0%	22	91.7%	24	88.9%	14	100.0%
	Total	150	100.0%	24	100.0%	27	100.0%	14	100.0%
PMS2	Loss of nuclear expression	11	7.3%	2	8.3%	2	7.4%	0	0.0%
	Intact nuclear expression	139	92.7%	22	91.7%	25	92.6%	14	100.0%
	Total	150	100.0%	24	100.0%	27	100.0%	14	100.0%

**Table 3 jpm-14-01152-t003:** Mismatch pepair (MMR) proteins by type of solid tumor.

	MSH 2				MSH6				MLH1				PMS2			
	Loss N.E. ^a^		Intact N.E. ^b^		Loss N.E. ^a^		Intact N.E. ^b^		Loss N.E. ^a^		Intact N.E. ^b^		Loss N.E. ^a^		Intact N.E. ^b^	
	*n*	*n* (%)	*n*	*n* (%)	*n*	*n* (%)	*n*	*n* (%)	*n*	*n* (%)	*n*	*n* (%)	*n*	*n* (%)	*n*	*n* (%)
Colorectal	3	2.0%	147	98.0%	7	4.7%	143	95.3%	15	10.0%	135	90.0%	11	7.3%	139	92.7%
Gastric	0	0.0%	24	100.0%	0	0.0%	24	100.0%	2	8.3%	22	91.7%	2	8.3%	22	91.7%
Endometrial	3	11.1%	24	88.9%	3	11.1%	24	88.9%	3	11.1%	24	88.9%	2	7.4%	25	92.6%
Esophageal	0	0.0%	14	100.0%	0	0.0%	14	100.0%	0	0.0%	14	100.0%	0	0.0%	14	100.0%

^a^ Loss N.E. = Loss of nuclear expression. ^b^ intact N.E. = Intact Nuclear Expression.

**Table 4 jpm-14-01152-t004:** MMR mismatch repair.

			dMMR (Deficient)	pMMR (Proficient)	Total
	Colorectal	*n*	19	131	150
		*n* (%)	12.7%	87.3%	100.0%
	Gastric	*n*	2	22	24
		*n* (%)	8.3%	91.7%	100.0%
	Endometrial	*n*	5	22	27
		*n* (%)	18.5%	81.5%	100.0%
	Esophageal	*n*	0	14	14
		*n* (%)	0.0%	100.0%	100.0%
Total		n	26	189	215
		*n* (%)	12.1%	87.9%	100.0%

**Table 5 jpm-14-01152-t005:** MMR mismatch repair by gender.

				dMMR (Deficient)	pMMR (Proficient)	Total
Female		Colorectal	*n*	5	55	60
			*n* (%)	8.3%	91.7%	100.0%
		Gastric	*n*	1	9	10
			*n* (%)	10.0%	90.0%	100.0%
		Endometrial	*n*	5	22	27
			*n* (%)	18.5%	81.5%	100.0%
		Esophageal	*n*	0	2	2
			*n* (%)	0.0%	100.0%	100.0%
	Total		*n*	11	88	99
			*n* (%)	11.1%	88.9%	100.0%
Male		Colorectal	*n*	14	76	90
			*n* (%)	15.6%	84.4%	100.0%
		Gastric	*n*	1	13	14
			*n* (%)	7.1%	92.9%	100.0%
		Esophageal	*n*	0	12	12
			*n* (%)	0.0%	100.0%	100.0%
	Total		*n*	15	101	116
			*n* (%)	12.9%	87.1%	100.0%
	Grand total		*n*	26	189	215
			*n* (%)	12.1%	87.9%	100.0%

**Table 6 jpm-14-01152-t006:** MMR mismatch repair by the site of analysis.

Site of Analysis				dMMR (Deficient)	pMMR (Proficient)	Total
Primary tumor		Colorectal	*n*	17	117	134
			n (%)	12.7%	87.3%	100.0%
		Gastric	*n*	1	19	20
			n (%)	5.0%	95.0%	100.0%
		Endometrial	*n*	5	20	25
			*n* (%)	20.0%	80.0%	100.0%
		Esophageal	*n*	0	14	14
			*n* (%)	0.0%	100.0%	100.0%
	Total		*n*	23	170	193
			*n* (%)	11.9%	88.1%	100.0%
Metastasis		Colorectal	*n*	2	14	16
			*n* (%)	12.5%	87.5%	100.0%
		Gastric	*n*	1	3	4
			*n* (%)	25.0%	75.0%	100.0%
		Endometrial	*n*	0	2	2
			*n* (%)	0.0%	100.0%	100.0%
	Total		*n*	3	19	22
			*n* (%)	13.6%	86.4%	100.0%
	Grand total		*n*	26	189	215
			*n* (%)	12.1%	87.9%	100.0%

**Table 7 jpm-14-01152-t007:** MMR mismatch repair by age.

				dMMR (Deficient)	pMMR (Proficient)	Total
<30		Colorectal	*n*	1	2	3
			*n* (%)	33.3%	66.7%	100.0%
	Total		*n*	1	2	3
			*n* (%)	33.3%	66.7%	100.0%
31–40		Colorectal	*n*	1	8	9
			*n* (%)	11.1%	88.9%	100.0%
		Gastric	*n*	0	3	3
			*n* (%)	0.0%	100.0%	100.0%
		Esophageal	*n*	0	1	1
			*n* (%)	0.0%	100.0%	100.0%
	Total		N	1	12	13
			*n* (%)	7.7%	92.3%	100.0%
41–50		Colorectal	*n*	3	11	14
			*n* (%)	21.4%	78.6%	100.0%
		Endometrial	*n*	1	3	4
			*n* (%)	25.0%	75.0%	100.0%
		Esophageal	*n*	0	2	2
			*n* (%)	0.0%	100.0%	100.0%
	Total		*n*	4	16	20
			*n* (%)	20.0%	80.0%	100.0%
51–60		Colorectal	*n*	2	25	27
			*n* (%)	7.4%	92.6%	100.0%
		Gastric	n	2	5	7
			*n* (%)	28.6%	71.4%	100.0%
		Endometrial	*n*	1	6	7
			*n* (%)	14.3%	85.7%	100.0%
		Esophageal	*n*	0	7	7
			*n* (%)	0.0%	100.0%	100.0%
	Total		*n*	5	43	48
			*n* (%)	10.4%	89.6%	100.0%
61–70		Colorectal	*n*	5	36	41
			*n* (%)	12.2%	87.8%	100.0%
		Gastric	*n*	0	7	7
			*n* (%)	0.0%	100.0%	100.0%
		Endometrial	*n*	3	11	14
			*n* (%)	21.4%	78.6%	100.0%
		Esophageal	*n*	0	3	3
			*n* (%)	0.0%	100.0%	100.0%
	Total		*n*	8	57	65
			*n* (%)	12.3%	87.7%	100.0%
71–80		Colorectal	*n*	4	39	43
			*n* (%)	9.3%	90.7%	100.0%
		Gastric	*n*	0	4	4
			*n* (%)	0.0%	100.0%	100.0%
		Endometrial	*n*	0	2	2
			*n* (%)	0.0%	100.0%	100.0%
		Esophageal	*n*	0	1	1
			*n* (%)	0.0%	100.0%	100.0%
	Total		*n*	4	46	50
			*n* (%)	8.0%	92.0%	100.0%
81–90		Colorectal	*n*	3	7	10
			*n* (%)	30.0%	70.0%	100.0%
		Gastric	*n*	0	3	3
			*n* (%)	0.0%	100.0%	100.0%
	Total		*n*	3	10	13
			*n* (%)	23.1%	76.9%	100.0%
>91		Colorectal	*n*	0	3	3
			n (%)		100.0%	100.0%
	Total		*n*	0	3	3
			*n* (%)		100.0%	100.0%
	Grand total		*n*	26	189	215
			*n* (%)	12,1%	87.9%	100.0%

## Data Availability

The data generated in this study are available upon request from the corresponding author.

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
