# Peer review of "Prevalence of Mismatch Repair Deficiency in Advanced Solid Tumors (Colorectal Cancer and Non-Colorectal Cancer) in One Mexican Institution"

_jpm, 2024, doi:10.3390/jpm14121152_

Round 1

Reviewer 1 Report

Comments and Suggestions for Authors

I believe that this article does not have such a big impact in its current form to be published. Unfortunately, although the limitations are recognized by the authors, they seem quite important to me personally. The only information from a scientific point of view that we receive is strictly related to the analysis of a subgroup of patients from a single center and is ultimately a single parameter, namely only the prevalence of MSI-H/dMMR in patients with the 4 tumor locations solid.

Also, the discussions are quite incomplete; they do not include important references from the literature related to the subject of the article.

Some reference ideas could be:

https://ascopubs.org/doi/10.1200/PO.22.00179

https://onlinelibrary.wiley.com/doi/10.1155/2020/1807929

https://link.springer.com/article/10.1007/s10147-024-02518-y

https://jitc.bmj.com/content/10/Suppl_2/A34

At the same time, I recommend the authors do the discussions by comparing the analysis from the literature in the same way from a single center of the prevalence of MSI-H/dMMR (here including centers from Europe, not only from the American continents).

In conclusion, i think the article does not bring anything new to the field. To be published the authors should bring more clinical correlations and/or statistical analyses with the presented results.

Author Response

Comment 1: I believe that this article does not have such a big impact in its current form to be published. Unfortunately, although the limitations are recognized by the authors, they seem quite important to me personally. The only information from a scientific point of view that we receive is strictly related to the analysis of a subgroup of patients from a single center and is ultimately a single parameter, namely only the prevalence of MSI-H/dMMR in patients with the 4 tumor locations solid..

Answer to comment 1:

We appreciate the reviewer’s insights and acknowledge the concerns regarding the limitations of our study. However, we would like to emphasize several key points that underscore the significance and potential impact of our research in the context of the challenges faced in biomarker testing for dMMR (deficient mismatch repair) in Mexico and the broader Latin American region.

1. Context of Biomarker Testing in Latin America: The landscape of biomarker testing in Latin America, particularly in countries such as Mexico, is undergoing significant transformation. This transition is driven by various factors including improved healthcare capabilities, strategic resource allocation, and ongoing efforts to obtain reimbursement for treatments aimed at dMMR entities, especially within the public health sector. Our study was conducted during a pivotal period marked by these advancements, rendering our findings especially pertinent to the current discourse on biomarker testing in the region.

2. Expertise and Representation: The investigators of this study, Dr. Motola and Dr. Dorantes, are esteemed leaders in the field of dMMR testing and oncology treatment in Mexico. Dr. Motola has previously served as the principal investigator of the KEYNOTE-177 study in Mexico. His collaboration with Dr. Dorantes, who conducted testing on patients under Dr. Motola’s care, underscores a robust foundation of expertise in this domain. The “Medica Sur” center, where this research was conducted, is thus well-positioned as a representative site for the study of dMMR within the local population.

3. Importance of Local Data: There is a significant gap in the understanding of dMMR and MSI-H (microsatellite instability-high) status among oncological patients in Mexico, primarily due to the lack of routine testing. This absence of existing data underscores the critical need for studies like ours, which aim to characterize the molecular profiles of the local population. Such characterization is essential for regulatory authorities responsible for approving access to innovative therapies, including immunotherapy, and ultimately contributes to improved patient outcomes.

4. Initial Insights as a Catalyst for Change: Although our study encompasses a subgroup analysis from a single center, it offers valuable initial insights into the prevalence of dMMR and MSI-H status within specific tumor types. These findings may act as a catalyst for future research, public awareness initiatives, and health policy changes. Establishing a foundational understanding in this under-researched

area is essential, as it can pave the way for more comprehensive studies in the future.

5. Limitations in Global Literature: We appreciate the reviewer’s recommendations regarding the literature, which primarily includes studies from Asia and the USA. It is important to note that one of these references is a meta-analysis; however, the authors of that analysis themselves acknowledge the insufficient data to make broad generalizations. Additionally, the study by Tetsuya Ito, which examined a single center with a sample size of n=3,609, demonstrates that even larger cohort studies often face limitations concerning their applicability to diverse populations.

This highlights the critical need for localized research, such as ours, to fill regional gaps in knowledge regarding the prevalence of dMMR.

In conclusion, we hope the reviewer recognizes the potential of our study to make a meaningful contribution to the literature on dMMR and to help facilitate improved access to immunotherapy for affected patients in Mexico and beyond.

Comment 2: Also, the discussions are quite incomplete; they do not include important references from the literature related to the subject of the article.

Some reference ideas could be:

https://ascopubs.org/doi/10.1200/PO.22.00179

https://onlinelibrary.wiley.com/doi/10.1155/2020/1807929

https://link.springer.com/article/10.1007/s10147-024-02518-y https://jitc.bmj.com/content/10/Suppl_2/A34

At the same time, I recommend the authors do the discussions by comparing the analysis from the literature in the same way from a single center of the prevalence of MSI-H/dMMR (here including centers from Europe, not only from the American continents).

Answer to comment 2: we restructured the discussion taking into consideration your kindly suggestion, some of the changes are, we include the points in the recommended literature, but additionally we highlighted the lack of Latin-American data.

Reviewer 2 Report

Comments and Suggestions for Authors

"Prevalence of Mismatch Repair Deficiency in Advanced Solid Tumors Colorectal Cancer and Non Colorectal Cancer ) in One Mexican Institution" is a descriptive study about the Mismatch Repair status in a heterogenous series of neoplasms, evaluated by immunohistochemistry.

The topic is interesting, and the manuscript is well written. I have few observations:

1) methods: the Authors should state clearly how many cases were retrospectively included in the study.
2) methods: more details about immunohistoxhemistry are needed: was IHC performed automatically) which platform was used? How the Authors defined the IHC results (definitions of proficient and deficient classes; patchy/low class?).
3) discussion: the authors say that there is work in which the prevalence of MSI is lower than that generally reported in large bowel carcinoma. The authors should refer to at least one further work with similar conclusions (Lieto E, Cardella F, Wang D, et al. Assessment of the DNA Mismatch Repair System Is Crucial in Colorectal Cancers Necessitating Adjuvant Treatment: A Propensity Score-Matched and Win Ratio Analysis. 
Cancers (Basel). 2023;16(1):134. Published 2023 Dec 27. doi:10.3390/cancers16010134).

4) discussion: patchy/low class is not present in the manuscript. Was immunohistochemistry sufficient in all cases to establish certainty the case as deficient of proficient?

Author Response

Comment 1:methods: the Authors should state clearly how many cases were retrospectively included in the study.

Answer 1: Change accepted, we add the statement in line 92 and 95 to clarify the total numbers of patients included.

2) methods: more details about immunohistochemistry are needed: was IHC performed automatically) which platform was used? How the Authors defined the IHC results (definitions of proficient and deficient classes; patchy/low class?).

Answer 2: Change accepted we add the text “Formalin-fixed and paraffin-embedded whole tissue sections, each measuring 4 µm in thickness, were prepared for immunohistochemical analysis. The immunohistochemistry (IHC) was performed using the following antibodies: MLH1 (G168-728), MSH2 (G219-1129), MSH6 (clone EP49), and PMS2 (EP51) Immunostains were performed using the standard avidin-biotin peroxidase method described in table 1, all were studied with appropriate controls. The control tissue used was normal colon.

Table 1. antibodies and dilutions used in the immunohistochemistry assay.

The resulting immunohistochemical slides were evaluated by two independent pathologists, who were blinded to each other’s assessments and had no prior knowledge of the clinicopathological parameters. Additionally, a third pathologist reviewed the pathology reports. In cases of discrepancies (which occurred in less than 5% of instances), a joint observation was conducted to reach a conclusive agreement.

Loss of MMR (mismatch repair) protein expression was defined by the absence of immunohistochemical staining in the nuclei of neoplastic cells. Cases were stratified into two distinct groups based on MMR protein expression following the recommendations of College of American Pathologist (CAP):

• Group 1: Proficient MMR (pMMR): Defined as cases with retained expression of all four MMR proteins.

• Group 2: Deficient MMR (dMMR): Defined as cases with loss of expression of one or more proteins, specifically within the MLH1/PMS2 and/or MSH2/MSH6 pairs. The classification of (iii) reduced MMR (low-patchy MMR) was not considered in this analysis, considering this classification as no widely accepted by CAP (14). In lines 119-142. It is important to highlight we do not use the patchy low classifications because is not recommended by College of American Pathologist.

3) discussion: the authors say that there is work in which the prevalence of MSI is lower than that generally reported in large bowel carcinoma. The authors should refer to at least one further work

with similar conclusions (Lieto E, Cardella F, Wang D, et al. Assessment of the DNA Mismatch Repair System Is Crucial in Colorectal Cancers Necessitating Adjuvant Treatment: A Propensity Score-Matched and Win Ratio Analysis. Cancers (Basel). 2023;16(1):134. Published 2023 Dec 27. doi:10.3390/cancers16010134).

Answer 3: change accepted we have modified the discussion structure and add to it some additional references.

Round 2

Reviewer 1 Report

Comments and Suggestions for Authors

I really appreciate your answers. Reading your comments and explanations really made me understand the necessity of publishing such an article and the importance and the impact it can still have despite this topic which is really quite discussed worldwide. 

Thus, I believe that this article deserves to be published precisely to be able to shed some light on real world data regarding patients from Mexico and those from Latin America and maybe even raise some awareness regarding the lack of testing.

Once again, thank you for the explanations provided.